# Physician Perceptions on Cancer Screening for LGBTQ+ Patients

**DOI:** 10.3390/cancers15113017

**Published:** 2023-06-01

**Authors:** Nicolas G. Nelson, Joseph F. Lombardo, Ayako Shimada, Marissa L. Ruggiero, Alexandria P. Smith, Kevin Ko, Amy E. Leader, Edith P. Mitchell, Nicole L. Simone

**Affiliations:** 1Department of Radiation Oncology, Sidney Kimmel Cancer Center, Thomas Jefferson University, Philadelphia, PA 19107, USA; 2Division of Biostatistics and Department of Experimental Pharmacology, Sidney Kimmel Cancer Center, Thomas Jefferson University, Philadelphia, PA 19107, USA; 3Department of Medical Oncology, Sidney Kimmel Cancer Center, Thomas Jefferson University Hospital, Philadelphia, PA 19107, USA

**Keywords:** cancer disparities, cancer screening, gender, sexual and gender minorities, structural and social determinants, LGBTQ+

## Abstract

**Simple Summary:**

Because systemic barriers contribute to the cancer disparities seen among LGBTQ+ patients, we asked physicians about their perceptions on cancer screenings for different subpopulations of this community. We also asked how many of these physicians had received LGBTQ+-specific training, whether they acknowledged that a patient’s LGBTQ+ status can affect their health needs, whether they felt confident in understanding these patients’ concerns, and whether they would feel comfortable being listed as an LGBTQ+-friendly practice. We also looked for relationships between their responses to certain questions and their self-reported gender, medical specialty, geographic location, and number of years of experience. The survey responses show a lack of agreement among physicians, a general willingness to learn how to better serve LGBTQ+ patients, and likely benefits of LGBTQ+-related training for physicians. Based on our results, we discuss potential areas for improvement in healthcare delivery and future research, including the need for clearer cancer screening standards for LGBTQ+ subpopulations.

**Abstract:**

The LGBTQ+ community experiences cancer disparities due to increased risk factors and lower screening rates, attributable to health literacy gaps and systemic barriers. We sought to understand the experiences, perceptions, and knowledge base of healthcare providers regarding cancer screening for LGBTQ+ patients. A 20-item IRB-approved survey was distributed to physicians through professional organizations. The survey assessed experiences and education regarding the LGBTQ+ community and perceptions of patient concerns with different cancer screenings on a 5-point Likert scale. Complete responses were collected from 355 providers. Only 100 (28%) reported past LGBTQ+-related training and were more likely to be female (*p* = 0.020), have under ten years of practice (*p* = 0.014), or practice family/internal medicine (*p* < 0.001). Most (85%) recognized that LGBTQ+ subpopulations experience nuanced health issues, but only 46% confidently understood them, and 71% agreed their clinics would benefit from training. Family/internal medicine practitioners affirmed the clinical relevance of patients’ sexual orientation (94%; 62% for medical/radiation oncology). Prior training affected belief in the importance of sexual orientation (*p* < 0.001), confidence in understanding LGBTQ+ health concerns (*p* < 0.001), and willingness to be listed as “LGBTQ+-friendly” (*p* = 0.005). Our study suggests that despite a paucity of formal training, most providers acknowledge that LGBTQ+ patients have unique health needs. Respondents had a lack of consensus regarding cancer screenings for lesbian and transgender patients, indicating the need for clearer screening standards for LGBTQ+ subpopulations and educational programs for providers.

## 1. Introduction

Lesbian, gay, bisexual, transgender, queer, and other sexual and gender minority (LGBTQ+) people face many disparities in healthcare, which are unfortunately associated with worse disease outcomes, leading the National Cancer Institute (NCI) to designate this population as medically underserved [1,2,3,4]. Bias in healthcare is known to contribute to the health disparities noted in the LGBTQ+ population [5], and despite increased cancer risks among sexual and gender minority populations, those who identify as LGBTQ+ are up to 25% less likely to undergo cancer screenings, such as colonoscopies, Pap smears, and mammograms, compared to their cisgender and straight counterparts [6,7]. Lack of cancer screening in this population is especially concerning given the elevated rates of malignancy and increased cancer risk factors including rates of smoking, alcohol intake, HPV (human papilloma virus) infection, poor diet, and hormone therapy [8]. The etiology of the lower rates of cancer screening for the LGBTQ+ population are likely multifactorial, stemming from community members’ experience with emotional distress from discrimination in the healthcare system, financial stressors, and a lack of cancer health knowledge [1,4,9,10]. Furthermore, as is often the case in nations with more severe social and legal anti-LGBTQ+ prejudice, patients can avoid seeking preventive care or medical attention altogether for fear of revealing their sexual orientation or gender identity and receiving hostility or backlash [6].

The exact extent to which the consistency of cancer screening recommendations by a healthcare provider affects patient uptake of cancer screening is unknown, but provider recommendations are a critical impetus [11]. In particular, the lack of medical provider experience or knowledge regarding health issues specific to the LGBTQ+ population has been cited as a reason that some community members do not see physicians or follow recommendations for routine health maintenance [12]. Cancer screening or outcome disparities may be exacerbated if providers are uncertain how LGBTQ+ status affects the care individual patients need, including cancer screening [10]. The reason for potential disparity in the recommendations from medical providers is certainly multifactorial and may include heteronormative assumptions, lack of LGBTQ+-specific information, or in some cases homophobia and transphobia [6,13]. The lack of national medical community consensus guidelines for cancer screening in the LGBTQ+ population contributes to this problem [1,14].

Cancer screening for LGBTQ+ patients should take into account the cancer risk factors known to be increased in this population [1,15,16]. In addition, each subpopulation’s risk factors must be accounted for; for example, the hormone replacement therapy and gender-affirming surgeries of varying types and schedules in the transgender population [6] and differences in sex development and baseline cancer risks in the intersex population [17]. It is imperative that providers be sensitive and knowledgeable to the LGBTQ+ status of patients in order to optimize care [13,18].

Since LGBTQ+ patients have risk factors for developing cancer but lower cancer screening rates, we sought to determine if healthcare providers understood the nuances of cancer screening recommendations for this community. We designed a survey to understand if there was consensus among providers on their perspectives of cancer screening for LGBTQ+ patients. Because our previous LGBTQ+ community-facing study suggested that apprehension about physician attitudes and beliefs may contribute to lower cancer screening rates [10], we gathered data from different specialists—those most responsible for recommending screenings, as well as others who may interface with patients in clinical or nonclinical settings. In addition, we sought to identify relationships between healthcare providers’ beliefs about the specific health needs and concerns of the LGBTQ+ community and their region of practice; level of training, i.e., years of clinical experience; and prior education in LGBTQ+-specific cultural competence.

## 2. Materials and Methods

An IRB-approved 20-item survey was created to assess the experience and perceptions of medical providers regarding cancer screenings in the LGBTQ+ community (Table A1, Appendix A). The survey anonymously asked 20 questions to assess physicians’:Prior medical experience and education specific to the LGBTQ+ community;Confidence in knowledge of patient concerns with screening using mammograms, Pap smears, and other HPV screening;Provider demographics including their medical specialty and geographic location.

Survey questions were answered using a 5-point Likert scale from “Strongly Disagree” to “Strongly Agree”, with “Neutral” in the middle. These included questions about providers’ confidence understanding the various health concerns of the LGBTQ+ community; their past experience with cultural competence training; their opinion of the importance of knowing patients’ sexual or gender minority status; and their clinics’ staff’s level of comfort serving LGBTQ+ patients (front desk staff, medical assistants, nurses and nurse practitioners, and technologists). In addition, specific questions about LGBTQ+-affirming aspects of their clinic were asked to determine how welcoming their practice is with respect to signage, gowns, and forms, as well as general beliefs about the LGBTQ+ population, e.g., whether they have specific health needs. The survey assessed if providers believed certain members of the LGBTQ+ population (i.e., lesbian, gay, and transgender men and women) had concerns with different cancer screenings (i.e., mammograms, Papanicolaou “Pap” smears, and other HPV screening including oral cancer screening). Other potentially sensitive cancer screenings including colonoscopy and PSA screening were omitted in order to limit the length of the survey.

Demographic characteristics of respondents were also evaluated with the respondents’ gender, number of years in practice (0–5, 6–10, or 11+), medical specialty, and the setting (rural, suburban, or urban) and geographical region of the United States where they practice. Estimated LGBTQ+ patient volumes were asked in order to approximate the number of LGBTQ+ community member patients each provider saw in practice within one month in closed-ended brackets. The survey ended with optional, open-ended questions to solicit recommendations for improving clinical encounters for LGBTQ+ patients (Table A1, results not presented). The electronic survey was distributed to physicians via RedCAP through the American College of Obstetricians and Gynecologists (ACOG), American Society for Radiation Oncology (ASTRO), American Urological Association (AUA), Alliance for Clinical Trials in Oncology (ACOSOG), and the Radiological Society of North America (RSNA). Participants were not compensated. All medical providers who completed the survey questions were included in the study.

Responses were aggregated into frequencies and percentages. Associations between providers’ medical specialty, length of time in practice, prior LGBTQ+-specific education, and beliefs about differential health and cancer screening issues among subpopulations of the LGBTQ+ community (i.e., whether subpopulations experience different health issues and whether lesbian, and transgender male and female patients have concerns with mammograms, Pap smears, and other HPV screening) were performed. For statistical purposes, we grouped medical and radiation oncologists (henceforth “oncologists”) and family and internal medicine specialists. Subgroup analysis was performed to determine if provider location, experience level, gender, and specialty influenced likelihood of prior formal LGBTQ+ education, current beliefs regarding the importance of knowing a patient’s sexual orientation to provide the best care, and opinions on whether their clinical staff would benefit from education about LGBTQ+ health issues.

To further analyze potential associations between provider characteristics and the consensus of their perspectives on different cancer screenings, we performed nine logistic regression analyses (three types of cancer screening for three patient populations) on possible contributing factors: providers’ years of practice, LGBTQ+-related training, medical specialty, and self-reported gender.

Key differences in responses based on different respondent categories were assessed using chi-square or Fisher’s exact tests with a significance level set to 0.05. All analyses were performed using SAS 9.4 (SAS Institute Inc., Cary, NC, USA). 

## 3. Results

From August 2017 to August 2018, 361 individuals responded to the survey with complete data sets available for 355 respondents. For subgroup analysis to determine if gender influenced the survey responses, the only respondent who reported a gender other than female or male was excluded from the analysis. 

### 3.1. Provider Demographics, Medical Specialty, and Geographic Location

Of the 355 respondents, 54% were female and 45% male. Slightly more than half of the healthcare providers had over ten years of experience (59%) and practiced in an urban setting (56%), but the geographic location of the healthcare providers was diverse. The location of each practitioner was broadly grouped as either coastal (Northeast or West) or noncoastal (Midwest or South). This geographic grouping was performed in an attempt to group geographic areas with more or less LGBTQ+-friendly policies and legislation [19]. Respondents represented diverse specialties including radiology (43%), internal medicine (15%), oncology (22.5%), and family medicine (12.1%). A summary of these provider characteristics is shown in Table 1.

### 3.2. Prior Experience and Education with the LGBTQ+ Community

Overall, 13% of the healthcare providers reported seeing less than one LGBTQ+ patient per month, 53% reported seeing one to five, and 14% reported seeing six to ten LGBTQ+ patients. This patient experience was similar for providers independent of the region they practice medicine. When asked about their confidence in understanding the various health concerns unique to the LGBTQ+ community, 47.6% agreed (or strongly agreed, henceforth “agreed”) that they were confident, which was also independent of geographic region of practice (Table A2). Despite varying comfort levels, 76.3% of healthcare providers agreed that they would want to be included in a list of “LGBTQ+-friendly” practices that would be accessible online or in an LGBTQ+ publication.

Overall, only 28% of respondents agreed with having received formal training regarding culturally competent interactions with LGBTQ+ patients. Notably, female providers were more likely to agree than male providers (34% vs. 22%, *p* = 0.020). Providers who had >10 years of practice were more likely to disagree that they had received formal training (*p* = 0.014, Figure A1). Family or internal medicine providers were more likely to agree with having undergone formal LGBTQ+ training (39%) as compared to oncology (28%) and radiology (22%, Figure A2). Of note, 71.3% of all respondents agreed that their clinics would benefit from LGBTQ+-specific training, and this was independent of length of time in training, region of practice, and gender of respondent.

When asked if it was important to know the sexual orientation of patients to provide the best care, 54.5% of healthcare providers agreed. Although this response did not vary by the gender of the respondent, more than 60% of providers who were in practice between 0–10 years agreed, with fewer (48.6%) providers agreeing who were in practice more than 10 years (27.9%). Family practice providers were more likely to agree with the importance of knowing their patients’ sexual orientation to provide the best care (94%), compared to oncology (62%) and radiology (22%). Conversely, 47% of radiologists, 37% of oncologists, and 6% of family and internal medicine providers did not believe that knowing their patients’ sexual orientation was required to deliver optimal care (*p* < 0.001, Figure 1).

When assessing specifically for the effect of providers’ experience on their likelihood of prior LGBTQ+-related training and their beliefs about LGBTQ+ patient care, significant differences based on number of years in practice were only seen regarding prior training (*p* = 0.014) with a trend for one belief (*p* = 0.053), which was the importance of knowing a patient’s sexual orientation to provide the best care (Table A3). More years in practice resulted in a lower likelihood of having received training, as well as lower agreement (and higher disagreement) with the clinical relevance of knowing patients’ sexual orientation.

Table 2 summarizes an assessment relating providers’ prior LGBTQ+-related training and their beliefs about LGBTQ+ patient care. Significant differences based on agreeing or disagreeing with having had training were seen for the importance of knowing a patient’s sexual orientation (*p* < 0.001), confidence in understanding LGBTQ+ health concerns (*p* < 0.001), and willingness to be listed as “LGBTQ+-friendly” (*p* = 0.005), but not the belief in unique health issues for LGBTQ+ subpopulations (92% if trained, 86% if untrained, *p* = 0.171). Trained physicians were significantly more likely to agree with the clinical importance of knowing the sexual orientation of their patients, their own confidence in understanding LGBTQ+ health concerns, and their willingness to be listed as an “LGBTQ+-friendly” practice.

Other survey questions regarded providers’ perceptions of specific LGBTQ+-relevant elements of their own clinics (Table A1, Questions 4–7); these responses are beyond the scope of this paper, as well as the respondents’ open-ended suggestions for improving these environments (Table A1, Questions 14–15).

### 3.3. Knowledge of Patient Concerns and Cancer Screening Using Mammograms, Pap Smears, and Other HPV Screening

To assess healthcare providers’ comfort with cancer screening in LGBTQ+ community members, specific questions were asked about screening for subpopulations of the community. Table 3 summarizes responses based on provider specialty. Overall, 85.4% of all respondents agreed that “within the LGBTQ+ community, subpopulations have very different health issues”. Family medicine practitioners almost unanimously agreed, showing a 96.9% consensus. In fact, family medicine, internal medicine, and oncology providers were all likely to agree that subpopulations in an LGBTQ+ community have distinct health issues, albeit at different rates (Table 3, Figure A3).

Providers disagreed about which cancer screenings are needed for different LGBTQ+ subpopulations, such as lesbian women, male-to-female (MTF) transgender women, and female-to-male (FTM) transgender men. Although most healthcare providers agreed that lesbian patients have concerns with mammogram screening, the rate of consensus varied among providers of different medical specialties with 85% for family/internal medicine, 72% for oncology, and 62% for radiology (*p* = 0.010). Overall, 84.5% of respondents agreed that mammograms were of concern for female-to-male (FTM) transgender patients, while 75.2% agreed that mammograms were of concern for male-to-female (MTF) transgender patients. Oncologists were less likely than specialists in family medicine, internal medicine, or radiology to agree on MTF patients’ need for mammogram screenings (*p* = 0.027).

Overall, Pap smears were believed to be of concern for lesbian patients by 74.6% of physicians, and 79.7% for FTM and 36% for MTF transgender patients. There were also differences between specialties in their agreement on Pap smear screening concerns; radiologists were least likely to agree that lesbian patients are concerned with getting Pap smears (*p* = 0.002). There were also provider differences in their beliefs on lesbian and FTM transgender patients’ need for other HPV screening (Table 3).

When assessing specifically for the effect of providers’ experience on their perceptions of patient concerns with different cancer screenings, significant differences based on number of years in practice were seen regarding mammography (*p* = 0.006), Pap smear (*p* = 0.003), and other HPV screening (*p* = 0.008) for lesbian women and mammography (*p* = 0.037) for FTM transgender men (Table A4). In general, the longer a physician was in practice, the more likely they were to disagree with lesbian or transgender patients having concerns with different cancer screenings.

### 3.4. Logistic Regression of Characteristics on Provider Consensus

For the nine logistic regression analyses of provider characteristics (LGBTQ+ training, specialty, and gender) on their perspectives on patient concerns with different cancer screenings, significant results are presented here by patient population (lesbian, FTM, MTF), as well as in Table A5.

Regarding lesbian patients, providers with 0–5 years of practice had almost three times higher odds of agreeing with concerns with mammograms than those with more than 10 years in practice (OR = 2.90, 95%: 1.28, 6.59) when adjusting for other characteristics. They also had about 2.3 times higher odds of agreeing with concerns about Pap smears (OR = 2.25, 95%: 0.99, 5.15), and 2.5 times higher odds of agreeing with other HPV screening concerns (OR = 2.50, 95%: 1.05, 5.96) than providers with more than 10 years in practice. In addition, providers in family or internal medicine had double odds of agreeing with the mammography concerns of lesbian patients compared to radiologists (OR = 2.02, 95%: 0.98, 4.20), 2.8 times higher odds of agreeing with their concerns with Pap smears (OR = 2.81, 95%: 1.22, 6.49), and 2.5 times higher odds of agreeing with their concerns with other HPV screening (OR = 2.28, 95%: 1.01, 5.15). Oncologists had slightly higher odds of agreeing with lesbians’ concern of mammograms than radiologists (OR = 1.11, 95%: 0.57, 2.14).

Regarding FTM patients (transmen), providers with 0–5 years of practice tended to have approximately 3.7 times higher odds of agreeing with the concern of mammography than providers with more than 10 years in practice (OR = 3.67, 95%: 0.83, 16.26) when adjusting for other characteristics. Family or internal medicine providers had more than 6 times higher odds of agreeing with the FTM patients’ concerns with mammogram compared to radiologists (OR = 6.24, 95%: 1.33, 29.27). On the other hand, regarding MTF patients (transwomen), family or internal medicine providers had half the odds of agreeing with the concern of mammography compared to radiologists (OR = 0.50, 95%: 0.25, 0.98).

### 3.5. LGBTQ+ Education and Provider Perspectives

When assessing specifically for the effect of prior LGBTQ+-related education, significant differences based on agreement or disagreement with having had training were seen only for mammography (*p* = 0.036) for lesbian women and Pap smear (*p* = 0.037) for FTM transgender men (Table 4). In both cases, providers who had received training were more likely to agree with patients’ concerns with these cancer screenings. Taken together, Table 2 and Table 4 indicate that formal training on culturally competent interactions with LGBTQ+ patients significantly correlated with providers’:Perceived importance of their patients’ sexual orientation (*p* < 0.001);Confidence in understanding LGBTQ+ health concerns (*p* < 0.001);Willingness to be listed as an “LGBTQ+-friendly” practice (*p* = 0.005);Perceptions of lesbian patients’ concerns with mammography (*p* = 0.036);Perceptions of transgender FTM patients’ concerns with Pap smear (*p* = 0.037).

## 4. Discussion

The data presented here demonstrate that a modest majority of healthcare providers believe it is important to know the LGBTQ+ status of their patients to provide optimal recommendations for cancer screening. Although most respondents evaluate at least one patient a month from the LGBTQ+ community, less than half are confident that they understand the unique health concerns of this population. Only 28% had had formal training on culturally competent interactions with LGBTQ+ patients, and most healthcare providers (71%) would like additional formal training. Providers had an understanding of concerns regarding cancer screening; however, a clear issue with nomenclature to help define cancer screening was noted, with differences between provider specialties. Although the reasons behind providers’ perspectives on LGBTQ+ health and cancer screening would require insights beyond the scope of our survey, this study provides several insights about the views and knowledgebase of providers regarding LGBTQ+ patients, which might be used to inform strategies for provider education and, ultimately, may help increase the uptake of appropriate cancer screenings in the LGBTQ+ population.

Overall, this study found that most providers did not have training on culturally sensitive interactions with LGBTQ+ patients but would like formal training. Prior studies demonstrate that LGBTQ+ patients’ fear of discrimination in the healthcare setting and lack of access to culturally competent healthcare are factors that decrease cancer screening and contribute to higher rates of cancer [20]. Our study is congruent with this. Only about a quarter of respondents to this survey reported that they had received at least some formal training, and those who had tended to be the physicians with fewer years in practice (Figure A1). Providers with >10 years of practice also trended toward disagreeing with the clinical importance of patients’ sexual orientation (*p* = 0.053), but on its own, years of experience did not significantly bear on providers’ confidence in understanding LGBTQ+ health issues (Table A3). The higher prevalence of LGBTQ+-specific training among newer providers we observed may owe in part to an increased prevalence of training and the availability of computer-based modules, which are particularly beneficial in areas where LGBTQ+ resources are scarcer [21]. Given that the likelihood of previous training was also specialty-dependent, it is likely that more institutions are now offering some form of LGBTQ+ training, but only for specific providers; however, our logistic regression analyses suggest that a provider’s number of years in practice and specialty can independently contribute to their perspectives on cancer screening among LGBTQ+ patients.

LGBTQ+-related training interventions have been implemented successfully, albeit sporadically in the US. Our study supports observations that relevant training helps providers feel more confident in understanding LGBTQ+ health needs and receptive to being included in LGBTQ+-friendly registries (Table 2). This underscores the notion that provider education can ultimately help reduce cancer disparities in multiple ways. For example, confidence in understanding health maintenance needs may motivate providers to prescribe cancer screenings more proactively, and increased public visibility of LGBTQ+-competent clinics may make community members more comfortable seeking preventive care. Educational training has been shown to improve providers’ knowledge and comfort levels regarding LGBTQ+ patient care and reduce explicit biases, even though these methods on their own were not shown to eliminate implicit biases [22]. For example, the pilot Health4LGBTI training course generally improved attitude scores for healthcare workers in Europe [23]. In Canada, a 90-min LGBTQ+ expert-led training on healthcare that discussed breast cancer in non-heterosexual women, and access and care for non-cisgender patients found significant improvements in LGBTQ+ health-specific knowledge and attitudes [24]. There are currently several working groups that could potentially help to address this disparity, including the Sexual and Gender Minority Research Working Group at the NIH and the ECOG-ACRIN Health Equity Committee [25,26]. Together, these results support the role of extending LGBTQ+-related education to more healthcare providers.

It is important to acknowledge that most (85%) respondents of our study agreed that subpopulations of the LGBTQ+ community experience distinct health issues and concerns, which was a highly prevalent belief regardless of prior training. Although cancer care providers agree about the existence of the different healthcare needs of their LGBTQ+ patients, they differed in their opinion on whether these disparities are relevant to their specialties. Notably, 37% of oncologists did not agree that knowing their patients’ sexual orientation was necessary for providing the best care (Figure 1); in contrast, only 6% of family or internal medicine providers disagreed with the importance of knowing their patients’ sexual orientation in the pursuit of delivering optimal care. The importance of learning and documenting a patient’s sexual orientation and gender identity is advocated for by the recommendations of the National Academy of Medicine and the Joint Commission [27,28], and our results suggest that prior training is significantly associated with appreciating the clinical relevance of one’s sexual orientation. Because respondents with LGBTQ+-related training might have sought it voluntarily due to preexisting beliefs, ongoing research on the efficacy of different forms of education is warranted.

Cancer screening for the LGBTQ+ population relies on provider knowledge of culturally sensitive terminology to recommend proper cancer screening, and this survey found notable issues with provider nomenclature knowledge and the importance placed on knowledge of LGBTQ+ status. Thirty-six percent of providers indicated Pap smears were a concern of transgender women, who do not have a cervix (Table 3, Table 4 and Table A4). This may be due to survey fatigue or difficulty understanding these questions, which could explain the larger proportion of unanswered questions related to cancer screenings; however, in part it could be due to providers’ unfamiliarity with LGBTQ+ nomenclature. Prior studies have shown that despite best intentions, provider misconceptions or perceived misconceptions significantly impact patients’ experience and quality of care [29,30], and further arguments for keeping abreast of appropriate terminology regarding LGBTQ+ communities have been enumerated [31]. In addition, 30% suggested FTM transgender men do not have concerns with breast cancer screening with mammograms. One limitation of this study is that questions regarding mammograms did not specify gender-affirming modalities such as top surgery or hormone therapy. Nuances regarding the type of top surgery to determine how much breast tissue is left and what imaging is indicated and length of time of hormone therapy is crucial to cancer screening recommendations. For example, breast cancer risk (and indication for mammography) is expected to vary, e.g., depending on whether breast tissue has been removed in transgender men, or perhaps on the duration of feminizing hormone therapy in transgender women. A Dutch study found that 60% of transgender women were prone to denser breast tissue on radiography, which is an independent risk factor for breast cancer; retrospectively, they did not experience an increased incidence, but the duration of feminizing hormone exposure should be considered in future studies [32]. Proper recommendations based on various clinical scenarios also highlight the importance of providers knowing the LGBTQ+ status of their patient. Once more, over a third of oncologists did not agree that knowing their patients’ sexual orientation was necessary for providing the best care (Figure 1). In our survey, 47% of radiologists disagreed that patients’ sexual orientation or gender identity is relevant to their role; however, data show that sexual orientation may affect the pretest probability of breast cancer in lesbian women [33], possibly due to a higher prevalence of risk factors including nulliparity [25,26]. The importance of learning and documenting a patient’s sexual orientation and gender identity is advocated for by the recommendations of the National Academy of Medicine and the Joint Commission [27,28]. Therefore, LGBTQ+ status can provide useful clinical context.

There was a general lack of consensus among providers regarding cancer screenings for lesbian and transgender patients in our study, demonstrating a need for clarity in cancer screening standards for LGBTQ+ subpopulations (Table 3). For instance, specialty-specific lack of consensus on whether patients had concerns with mammography existed regarding transgender women (*p* = 0.009), transgender men (*p* = 0.027), and lesbian women (*p* = 0.010). Notably, accordance with the statement that LGBTQ+ subpopulations face unique health issues was specialty-dependent (*p* < 0.001) and not affected by prior training (*p* = 0.171, Table 2). Because of this lack of consistency in perceptions of cancer screenings for transgender patients, which can be affected by training (Table 4), we would suggest that any sexual and gender minority cultural competence training should include training specific to transgender patients across the spectra of gender transitions.

There are other sensitive cancer screenings on which physicians should be prepared to provide LGBTQ+ patients with clinically and culturally sensitive advice, e.g., colonoscopy and prostate-specific antigen (PSA) screening for anal and prostate cancers, respectively. Efforts are underway to consolidate and reconcile existing data into actionable guidelines, though they have yet to approach any widely accepted agreement [34]. Future initiatives could include LGBTQ+-related training for cancer care providers as discussed previously, including education about LGBTQ+ terminology, cultural sensitivity for sexual and gender minorities, specific health concerns and risk factors associated with different subpopulations of the LGBTQ+ community, and medical and psychosocial considerations for patients on different spectra of gender transition. This training would increase providers’ ability and confidence not only in caring for their LGBTQ+ patients, but also in educating their patients, peers, and future providers [35].

Some institutions are working to incorporate sexual orientation and gender identity (SOGI) data in their electronic medical records [36], which is useful for researching disparities and informing preventive health strategies, and SOGI collection is often the first step in addressing health and social issues related to the LGBTQ+ status of patients [36]. Although the responses to the survey’s open-ended questions about improving patient encounters for LGBTQ+ patients are beyond the scope of this paper, collecting SOGI data was a frequent suggestion, along with proper training for providers and staff. In addition to formal LGBTQ+-related training and patient SOGI collection, we further advocate consensus conferences to disseminate known statistics, generate working recommendations, and form hypotheses working toward national standards of cancer screening practices specific to the LGBTQ+ population.

Finally, this survey does not take into account when, why, or how formal LGBTQ+-related training was obtained by those who received it, but the results do support an association between provider education and their attitudes, beliefs, and knowledge base regarding LGBTQ+ patients. Formal training on culturally competent interactions with LGBTQ+ patients significantly correlated with providers’ belief in the importance of their patients’ sexual orientation, confidence in understanding LGBTQ+ health concerns, willingness to be listed as an “LGBTQ+-friendly” practice, and perceptions of lesbian patients’ concerns with mammography and FTM patients’ concerns with Pap smear. These data demonstrate that cultural competency training may result in proactive cancer screening referrals, visibly inclusive care options, and ultimately a reduction in cancer disparities among LGBTQ+ patients.

## 5. Conclusions

Healthcare providers play a crucial role in ensuring compliance of their patients with cancer screening. Despite the identified increased cancer risk factors and decreased cancer outcomes in the LGBTQ+ community, community members have lower rates of cancer screening which may in part be due to the lack of consensus among healthcare providers for what screening should be performed for which patients.

Our study found that providers are overall sympathetic to the unique health needs of LGBTQ+ patients and are willing to learn how to better serve them, but they demonstrated a lack of consensus regarding recommendations for cancer screening. Formal LGBTQ+ cultural competency training portended more confidence in delineating screening, as well as willingness to demonstrate allyship as an “LGBTQ+-friendly” practice.

The lack of consensus regarding specific cancer screenings for LGBTQ+ patients may relate to the general lack of formal training on LGBTQ+ specific health issues observed, but this gap may be narrowing over time as providers with fewer years in practice were more likely to report formal LGBTQ+ cultural competency training. Altogether, these results highlight the need for:National efforts toward a consensus on systematic SOGI collection and clear cancer screening standards for LGBTQ+ subpopulations;Improved educational programs for medical providers in primary care, oncology, and beyond, with the goals of improving healthcare delivery for the LGBTQ+ community and reducing disparities.

## Figures and Tables

**Figure 1 cancers-15-03017-f001:**
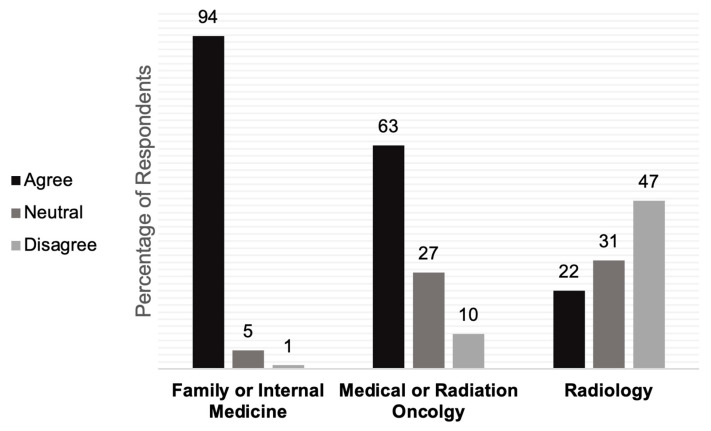
Provider specialty correlates with beliefs about the importance of patient LGBTQ+ status. Subgroup analysis by specialty, proportional responses to the question: “It is important to know the sexual orientation of my patients to provide the best care”, *p* < 0.001.

**Table 1 cancers-15-03017-t001:** Descriptive statistics of demographics for participating cancer providers completing the survey, n = 355.

Question	Response	All (n = 355)
Gender, n (%) ^1^	Female Male Other	192 (54.1%) 160 (45.1%) 1 (0.3%)
How many years have you been in practice? n (%)	0–5 6–10 >10	68 (19.2%) 77 (21.7%) 208 (58.6%)
Which of the following best describes your practice, n (%) ^2^	Family Medicine Gynecology Internal Medicine Medical Oncology Radiation Oncology Radiology Surgery Urology Other ^3^	43 (12.1%) 4 (1.1%) 53 (14.9%) 51 (14.4%) 30 (8.5%) 154 (43.4%) 7 (2.0%) 8 (2.3%) 5 (1.4%)
Which of the following best describes the area you practice in? n (%) ^2^	Rural Suburban Urban	34 (9.6%) 117 (33.0%) 200 (56.3%)
What region do you practice in? n (%) ^2^	Mid-Atlantic Midwest Northeast South West Other ^4^	26 (7.3%) 84 (23.7%) 87 (24.5%) 72 (20.3%) 77 (21.7%) 5 (1.4%)

^1^ There are 2 missing (0.6%). ^2^ There are 4 missing (1.1%). ^3^ Other category includes HIV primary care (1), palliative medicine/medical oncology (1), hematology/stem cell transplant (1), and other entries (2). ^4^ Other category includes Mexico (1) and Southwest (4).

**Table 2 cancers-15-03017-t002:** Descriptive statistics summary of binarized agreement with previous LGBTQ+ education and providers’ belief in LGBTQ+-specific health issues, perceived importance of knowing patients’ sexual orientation, confidence in understanding LGBTQ+ health concerns, and willingness to be listed “LGBTQ+-friendly” by the number of years in practice, n = 272. Of the 356 providers who responded to the question on previous LGBTQ+ training, 84 (23.6%) answered “Neutral” and were omitted here.

		Responded to “I Have Had Formal Training on Culturally-Competent Interactions with LGBTQ+ Patients”:		
Question	Response	Yes (Strongly Agree/ Agree, n = 100)	No (Disagree/ Strongly Disagree, n = 172)	*p*-Value
Within the LGBTQ+ community, subpopulations have very different health issues, n (%)	Strongly Agree/ Agree	92 (92.0%)	147 (85.5%)	0.171
	Neutral	5 (5.0%)	20 (11.6%)	
	Disagree/ Strongly Disagree	3 (3.0%)	5 (2.9%)	
It is important to know the sexual orientation of my patients to provide the best care, n (%)	Strongly Agree/ Agree	67 (67.0%)	74 (43.3%)	<0.001
	Neutral	21 (21.0%)	42 (24.6%)	
	Disagree/ Strongly Disagree	12 (12.0%)	55 (32.2%)	
I am confident I understand various health concerns unique to the LGBTQ+ community, n (%)	Strongly Agree/ Agree	74 (74.8%)	42 (24.6%)	<0.001
	Neutral	19 (19.2%)	75 (43.9%)	
	Disagree/ Strongly Disagree	6 (6.0%)	54 (31.6%)	
Given the option, I would want to be included in a list of “LGBTQ+-friendly” practices that would be accessible online and/or in an LGBTQ+ publication, n (%)	Strongly Agree/ Agree	88 (88.9%)	125 (73.1%)	0.005
	Neutral	7 (7.1%)	36 (21.1%)	
	Disagree/ Strongly Disagree	4 (4.0%)	10 (5.9%)	

**Table 3 cancers-15-03017-t003:** Descriptive statistics of questions regarding different LGBTQ+ patient health issues and concerns with cancer screenings by provider specialty, n = 355.

Question	Response	All (n = 355)	Family or Internal Medicine (n = 96)	Medical or Radiation Oncology (n = 82)	Radiology (n = 154)	Other (n = 23)	*p*-Value
Within the LGBTQ+ community, subpopulations have very different health issues, n (%)	Strongly Agree/Agree	303 (85.4%)	93 (96.9%)	73 (89.0%)	118 (76.6%)	19 (82.6%)	<0.001
	Neutral	42 (11.8%)	3 (3.1%)	7 (8.5%)	29 (18.8%)	3 (13.0%)	
	Disagree/Strongly Disagree	9 (2.5%)	0 (0.0%)	1 (1.2%)	7 (4.5%)	1 (4.3%)	
Lesbian patients have concerns with:							
Mammogram screening, n (%) ^1^	Strongly Agree/Agree	255 (71.8%)	82 (85.4%)	59 (72.0%)	95 (61.7%)	19 (82.6%)	0.010
	Disagree/Strongly Disagree	84 (23.7%)	14 (14.6%)	19 (23.2%)	47 (30.5%)	4 (17.4%)	
Pap smear screening, n (%) ^2^	Strongly Agree/Agree	265 (74.6%)	87 (90.6%)	63 (76.8%)	96 (62.3%)	19 (82.6%)	0.002
	Disagree/Strongly Disagree	68 (19.2%)	9 (9.4%)	15 (18.3%)	40 (26.0%)	4 (17.4%)	
Other HPV screening (oral health), n (%) ^3^	Strongly Agree/Agree	265 (74.6%)	86 (89.6%)	62 (75.6%)	98 (63.6%)	19 (82.6%)	0.010
	Disagree/Strongly Disagree	67 (18.9%)	10 (10.4%)	15 (18.3%)	38 (24.7%)	4 (17.4%)	
Female-to-male transgender patients [trans men] ^9^ have concerns with:							
Mammogram screening, n (%) ^4^	Strongly Agree/Agree	300 (84.5%)	93 (96.9%)	67 (81.7%)	120 (77.9%)	20 (87.0%)	0.009
	Disagree/Strongly Disagree	32 (9.0%)	2 (2.1%)	8 (9.8%)	19 (12.3%)	3 (13.0%)	
Pap smear screening, n (%) ^5^	Strongly Agree/Agree	283 (79.7%)	88 (91.7%)	68 (82.9%)	107 (69.5%)	20 (87.0%)	0.140
	Disagree/Strongly Disagree	38 (10.7%)	7 (7.3%)	7 (8.5%)	22 (14.3%)	2 (8.7%)	
Other HPV screening (oral health), n (%) ^6^	Strongly Agree/Agree	283 (79.7%)	88 (91.7%)	66 (80.5%)	109 (70.8%)	20 (87.0%)	0.315
	Disagree/Strongly Disagree	37 (10.4%)	7 (7.3%)	8 (9.8%)	20 (13.0%)	2 (8.7%)	
Male-to-female transgender patients [trans women] ^9^ have concerns with:							
Mammogram screening, n (%) ^7^	Strongly Agree/Agree	267 (75.2%)	83 (86.5%)	54 (65.9%)	115 (74.7%)	15 (65.2%)	0.027
	Disagree/Strongly Disagree	70 (19.7%)	13 (13.5%)	22 (26.8%)	27 (17.5%)	8 (34.8%)	
Pap smear screening, n (%) ^8^	Strongly Agree/Agree	128 (36.1%)	34 (35.4%)	32 (39.0%)	56 (36.4%)	6 (26.1%)	0.368
	Disagree/Strongly Disagree	194 (54.6%)	62 (64.6%)	42 (51.2%)	74 (48.1%)	16 (69.6%)	
Other HPV screening (oral health), n (%) ^8^	Strongly Agree/Agree	280 (78.9%)	86 (89.6%)	64 (78.0%)	110 (71.4%)	20 (87.0%)	0.729
	Disagree/Strongly Disagree	42 (11.8%)	10 (10.4%)	10 (12.2%)	20 (13.0%)	2 (8.7%)	

^1^ There are 16 (4.5%) missing: 4 in medical/radiation oncology, 12 in radiology. ^2^ There are 22 (6.2%) missing: 4 in medical/radiation oncology, 18 in radiology. ^3^ There are 23 (6.5%) missing: 5 in medical/radiation oncology, 18 in radiology. ^4^ There are 23 (6.5%) missing: 1 in family/internal medicine, 7 in medical/radiation oncology, 15 in radiology. ^5^ There are 34 (9.6%) missing: 1 in family/internal medicine, 1 in other, 7 in medical/radiation oncology, 25 in radiology. ^6^ There are 35 (9.9%) missing: 1 in family/internal medicine, 1 in other, 8 in medical/radiation oncology, 25 in radiology. ^7^ There are 18 (5.1%) missing: 6 in medical/radiation oncology, 12 in radiology. ^8^ There are 33 (9.3%) missing: 8 in medical/radiation oncology, 24 in radiology, 1 in other. ^9^ Bracketed descriptions were not included in survey.

**Table 4 cancers-15-03017-t004:** Descriptive statistics summary of binarized agreement with previous LGBTQ+ education and providers’ perceptions of concerns with cancer screenings, n = 272. Of the 356 providers who responded to the question on previous LGBTQ+ training, 84 (23.6%) answered “Neutral” and were omitted here.

		Responded to “I Have Had Formal Training on Culturally-Competent Interactions with LGBTQ+ Patients”:		
Question	Response	Yes (Strongly Agree/ Agree, n = 100)	No (Disagree/ Strongly Disagree, n = 172)	*p*-Value
Lesbian patients have concerns with:				
Mammogram screening, n (%) ^1^	Strongly Agree/ Agree	79 (80.6%)	112 (68.7%)	0.036
	Disagree/ Strongly Disagree	19 (19.4%)	51 (31.3%)	
Pap smear screening, n (%) ^2^	Strongly Agree/ Agree	81 (84.4%)	119 (74.8%)	0.073
	Disagree/ Strongly Disagree	15 (15.6%)	40 (25.2%)	
Other HPV screening (oral health), n (%) ^3^	Strongly Agree/ Agree	77 (81.1%)	119 (74.8%)	0.254
	Disagree/ Strongly Disagree	18 (19.0%)	40 (25.2%)	
Female-to-male transgender patients have concerns with:				
Mammogram screening, n (%) ^4^	Strongly Agree/ Agree	92 (92.9%)	142 (88.2%)	0.217
	Disagree/ Strongly Disagree	7 (7.1%)	19 (11.8%)	
Pap smear screening, n (%) ^5^	Strongly Agree/ Agree	90 (93.8%)	131 (85.1%)	0.037
	Disagree/ Strongly Disagree	6 (6.3%)	23 (14.9%)	
Other HPV screening (oral health), n (%) ^5^	Strongly Agree/ Agree	89 (92.7%)	130 (84.4%)	0.053
	Disagree/ Strongly Disagree	7 (7.3%)	24 (15.6%)	
Male-to-female transgender patients have concerns with:				
Mammogram screening, n (%) ^1^	Strongly Agree/ Agree	82 (82.8%)	122 (75.3%)	0.154
	Disagree/ Strongly Disagree	17 (17.2%)	40 (24.7%)	
Pap smear screening, n (%) ^6^	Strongly Agree/ Agree	36 (37.5%)	65 (42.8%)	0.411
	Disagree/ Strongly Disagree	60 (62.5%)	87 (57.2%)	
Other HPV screening (oral health), n (%) ^6^	Strongly Agree/ Agree	85 (88.5%)	127 (83.6%)	0.277
	Disagree/ Strongly Disagree	11 (11.5%)	25 (16.5%)	

^1^ There are 11 missing (4.0%). ^2^ There are 17 missing (6.3%). ^3^ There are 18 missing (6.6%). ^4^ There are 12 missing (4.4%). ^5^ There are 22 missing (8.1%). ^6^ There are 24 missing (8.8%).

## Data Availability

The data presented in this study are available in FigShare at https://doi.org/10.6084/m9.figshare.23271467, reference number [37].

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
