# Peer review of "Physician Perceptions on Cancer Screening for LGBTQ+ Patients"

_cancers, 2023, doi:10.3390/cancers15113017_

Round 1

Reviewer 1 Report

Firstly, I would like to thank you for inviting me to review the article entitled: Physician Perceptions on Cancer Screening for LGBTQ+ Patients.

The title accurately reflects the manuscript. The manuscript is well written in terms of clarity, style, and use of English and has a logical construction. The discussion section explains the case in the context of published information. The conclusions accurately and clearly explain the main message. The figures and tables are of good quality and relevant to the clinical message. The references are appropriate and current.

 I can find no major flaws in the manuscript. Its major strength is its originality as a subject.

 Minor comment:

1) On page 2, lines 57 and 58, the authors could add to the reasons for lower rates of cancer screening for LGBTQ+ that in some less developed countries, they have to hide their identity as LGBTQ+ because society is hostile against these people and avoid seeking medical attention.

Author Response

The authors appreciate the thorough review of this paper, and we are hope that it will be useful to the community.

Comment 1) The authors agree that this is an important point. Although the scope of our survey focused on United States providers, we have included this point with a reference in the Introduction as suggested because it is indeed relevant background information.

Reviewer 2 Report

The authors examined the physician’s perceptions on cancer screening for LGBTQ+ patients, more specifically, mammogram, Pap smear, and HPV screening for lesbian, female to transgender, and male to female transgender patients.  This work is important and potentially contributes to this under-studied area of cancer health disparities. The below is my comments:

This study is descriptive in the way that authors just report frequencies of answers by categories based on physicians’ medical specialties or years of experience.  Was there any specific hypothesis to test?  

What are the factors influencing the physicians’ perception about cancer screening among LGBTQ+ patients?

Perceptions of cancer screening among family or internal medicine physicians were compared to oncologists and radiologists.  I am not sure what is the clinical or public health significance of this comparison, because oncologists and radiologists are not the ones who are recommending mammography, pap smear, and HPV screening to LGBTQ+ patients.  Maybe, only physicians in the specialties that are involved in cancer screening recommendations should be included here?

The figures (bar plots) – There should be a label for y-axis in each figure.  Instead of the number of physicians in each category, the percentage might be more helpful, because the number of physicians who participated are different for each category of medical specialties.

There seems to be results of some survey questions that are not presented in this paper? 

I understand why the authors focus on mammography, pap smear, and HPV screening, but how about other cancer screening? Physicians need to be sensitive when they are recommending colonoscopy and PSA screening for LGBTQ+ patients.

Author Response

The authors appreciate the thoughtful review of our paper, and we have tried to address all comments as follows:

Comment 1) The authors agree that the nature and confidence threshold of the study must be included. Our hypothesis was that different categories of providers would tend to differ in their agreement with various statements about LGBTQ+ care and cancer screenings, set to a confidence value of p<0.05, which we have emphasized in the Methods.

Comment 2) The authors agree that this would be an important consideration, and we have commented on this limitation in the Discussion. To answer this question would require insights that go beyond the scope of the questions asked in this survey.

Comment 3) The authors agree that an explanation of the logic behind including non-primary care specialists deserves including, which we have incorporated into the Introduction. Essentially, our previous LGBTQ+ community-facing survey study (reference #10) suggested that many LGBTQ+ patients are apprehensive about their providers’ attitudes and beliefs, which may contribute to lower cancer screening rates. Our thought was that because of the multidisciplinary nature of medicine, it was relevant to know the opinion of different specialists—those most responsible for recommending screenings as well as others—because they are a part of the medical community with whom patients may interact, both in clinical settings and otherwise.

Comment 4) The authors agree that the y-axes warrant labeling, and we have included labels. While drafting, the authors initially considered including percent representation from each specialty (as opposed to numbers) in the bar graphs of Figure 3, but ultimately we wanted to generate a visual that more accurately reflected the body of survey responses as a whole and was consistent with the other bar graphs. In addition, the percentages are available in the text referencing the figure.

Comment 5) The authors acknowledge that responses to the open-ended questions were not presented in the results section because they extended beyond the study’s objective of gauging physician attitudes and beliefs (and not systematically collecting qualitative suggestions for improvement). We have now alluded to these responses in the Results and briefly referenced them in the Discussion.

Comment 6) The authors agree that this is a key point and a significant limitation due to the survey’s length, and we have mentioned these other screenings in the Discussion in addition to rationalizing their omission in the Methods.

Round 2

Reviewer 2 Report

This manuscript covers important issues related to the LGBTQ+ and cancer screening. There was a minimal improvement in manuscript from the original submission. More careful considerations for research design, analysis, and presentation are recommended. There should be important questions that can be addressed besides simply presenting the differences in awareness of cancer screening between medical specialties.  Having had a LGBTQ+ education and length of practice influence awareness of cancer screening in LGBTQ+ patients?  Are the differences in cancer screening awareness based on medical specialties significant accounting for LGBTQ+ education and year of practice?  Why is the difference important and how knowing these improve care?  Who should get more LGBTQ+ education?

Author Response

This manuscript covers important issues related to the LGBTQ+ and cancer screening. There was a minimal improvement in manuscript from the original submission. More careful considerations for research design, analysis, and presentation are recommended.

The authors are grateful for the thoughtful suggestions, which have resulted in significant improvements. For example, we have created four new tables with our statistician, and these tables have strengthened the manuscript’s conclusions. We have also added editorial in the Discussion commenting on this new content as well as clarifying previous points that were brought to our attention.

There should be important questions that can be addressed besides simply presenting the differences in awareness of cancer screening between medical specialties. Having had a LGBTQ+ education and length of practice influence awareness of cancer screening in LGBTQ+ patients?

We agree and have created new tables (3 and 4) to assess the effects of length of practice and LGBTQ+ education on provider beliefs and perceptions on cancer screenings. The results have significantly bolstered the manuscript’s conclusion, in particular illustrating the power of LGBTQ+-related training in various attitudes and beliefs but not necessarily awareness that unique health challenges exist.

Are the differences in cancer screening awareness based on medical specialties significant accounting for LGBTQ+ education and year of practice?

We have created new tables (6 and 7) with relevant commentary in the discussion to illustrate and consider associations between experience and education and differences in cancer screening perceptions.

Why is the difference important and how knowing these improve care? Who should get more LGBTQ+ education?

We have taken the new observations drawn from the expanded analyses described above and extended them to our Discussion and Conclusions.

Round 3

Reviewer 2 Report

Authors made some revisions to the manuscript.  Overall, only descriptive statistics and Chi-square tests were used in this study. What were major contributing factors for physicians’ perspectives on cancer screening among LGBTQ+ patients? Years of practice, having had LGPTQ+ education, and medical specialties may be all correlated in this dataset.  For example, family and internal medicine doctors in this study tend to have shorter years of practice and have had LGBTQ+ education?  Radiologists are more likely to have longer years of practice or less likely to have had LGBTQ+ education?  I can see clinicians who recently went through training may have had LGBTQ+ education as a part of their curriculum, but such training program was not widely available until recently.

A lot of analysis results were presented, but it is hard to go through them all switching back and forth between tables/figures and text.  Are all these tables necessary to be included in the main text?  Can some of them go into supplementary materials?  Also, discussion emphasizes on importance of LGBTQ+ education and training determine knowledge and perception on LGBTQ+ and reducing disparities.  If that was a take home point, maybe the results can be presented to illustrate the needs for LGBTQ+ education and training.

For bar graph, I still think % of respondents should be shown instead of number of respondents, because total number of respondents in each category (eg., clinical specialties) is different and it is hard to compare different categories.

Author Response

Thank you for these suggestions, we have now revised the manuscript to address these items.

Authors made some revisions to the manuscript.  Overall, only descriptive statistics and Chi-square tests were used in this study. What were major contributing factors for physicians’ perspectives on cancer screening among LGBTQ+ patients? Years of practice, having had LGPTQ+ education, and medical specialties may be all correlated in this dataset.  For example, family and internal medicine doctors in this study tend to have shorter years of practice and have had LGBTQ+ education?  Radiologists are more likely to have longer years of practice or less likely to have had LGBTQ+ education?  I can see clinicians who recently went through training may have had LGBTQ+ education as a part of their curriculum, but such training program was not widely available until recently.

First, to further analyze potential associations between provider characteristics and perspectives (i.e., consensus on different cancer screenings), we performed logistic regression on possible contributing factors: years of practice, having had a formal LGBTQ+ education, medical specialties, and providers’ gender, with results listed in Section 3.4. We have commented on potential confounding factors in the discussion, including the recency of more widespread LGBTQ+ training (i.e., more training among people who went through medical education more recently).

A lot of analysis results were presented, but it is hard to go through them all switching back and forth between tables/figures and text.  Are all these tables necessary to be included in the main text?  Can some of them go into supplementary materials?  Also, discussion emphasizes on importance of LGBTQ+ education and training determine knowledge and perception on LGBTQ+ and reducing disparities.  If that was a take home point, maybe the results can be presented to illustrate the needs for LGBTQ+ education and training.

Next, the authors agree that the previous version of the manuscript was inundated with tables and figures, so we have relegated selected tables and figures to the Appendix to avoid interrupting the paper’s narrative flow while also keeping the data available to readers. Namely, Section 3.2 now focuses on the significance of LGBTQ+-related training, and Section 3.3 highlights the lack of consensus on proper screening, which we discuss in the Discussion with respect to the need for consistent guidelines. To highlight the potential benefits of LGBTQ+ education and training, we have separated associations between LGBTQ+ education and specific responses into its own Section 3.5. Although indicating whether a provider’s perception of a particular cancer screening was “correct”/“incorrect” was beyond the scope of the survey study, in order to illustrate the role of LGBTQ+ training in affecting provider consensus, we have referred to the data illustrating each point in the Discussion.

For bar graph, I still think % of respondents should be shown instead of number of respondents, because total number of respondents in each category (eg., clinical specialties) is different and it is hard to compare different categories.

For the bar graphs, the authors have altered the aforementioned figures to illustrate percentages instead of number of respondents.